# Adversarial enhanced representation for link prediction in multi-layer networks

## Abstract

Multi-layer networks are widely utilized in various applications, including social networks, biological networks, and Internet typologies. In these networks, link prediction is a longstanding issue that predicts missing links based on the observed structures across all layers, thereby assisting in tasks such as network recovery and drug-target prediction. However, existing link prediction methods tend to learn nontransferable intra-layer representations that cannot generalize well to other layers, which results in inefficient utilization of the structural correlations between layers in multi-layer networks. To address this, we propose a novel graph embedding method called Adversarial Enhanced Representation ($AER$) for link prediction in multi-layer networks. $AER$ comprises three modules: a representation generator, a layer discriminator, and a link predictor. The representation generator is designed to learn and fuse the links' inter-layer and intra-layer representations. Also, the layer discriminator aims to identify the layer sources of learned inter-layer representations. During a minimax two-player game, the representation generator attempts to learn inter-layer transferable representations to deceive the layer discriminator. In order not to be deceived, the layer discriminator attempts to accurately distinguish the layer sources of learned inter-layer representations. Finally, the link predictor works in collaboration with the representation generator to predict whether a link is a missing link based on the adaptive fusion between inter-layer transferable and intra-layer representations. To validate the effectiveness of our proposed method, we conduct extensive experiments on real-world datasets. The experimental results demonstrate that $AER$ outperforms state-of-the-art methods in link prediction performance for multi-layer networks.

## 1 Introduction

In recent years, the analysis of complex networks has seen a surge in interest, leading to the popularity of the research line of multi-layer networks Poledna et al. (2015); Belyi et al. (2017); Lei et al. (2020). The structures in multi-layer networks that consist of different layers provide a more comprehensive representation of interactions and information exchanges across different sources Zhang et al. (2005). For example, people engage in various correlated forms of communication and interaction, such as friendships, virtual social relationships, and telephone communications. In multi-layer networks, the main task of link prediction is to predict missing links based on the observed structures across all layers Kumar et al. (2020). However, unlike traditional link prediction in single-layer networks, link prediction in multi-layer networks poses the challenge of effectively leveraging inter-layer and intra-layer structural informaiton to achieve accurate predictions.

Existing link prediction methods, including both topological calculation methods Samei & Jalili (2019); Nasiri et al. (2022) and deep learning methods Jalili et al. (2017); De Bacco et al. (2017), have shown impressive results. Topological calculation methods exploit the topological structures to predict the existing possibility of unobserved links between two known nodes Yao et al. (2017). On the other hand, deep learning methods have achieved impressive performance improvements by leveraging their superior feature extraction capabilities Shan et al. (2020). However, these existing methods often focus on the topology structures within individual layers and tend to learn nontransferable intra-layer representations that cannot generalize well to other layers. They cannot efficiently utilize the structural correlations between layers in multi-layer networks. They primarily learn intra-layer representations for link prediction on each layer, neglecting the potential benefits of inter-layer

representations from other layers. This limitation significantly restricts the link prediction performance in multi-layer networks. Although the interactions between nodes may vary across layers, nodes that are the same in different layers often exhibit similar preferences or patterns in link generation Bai et al. (2021). For instance, if two people are friends in one layer, there is a high likelihood that they will engage in conversations in another layer. Although the intra-layer representations in different layers have some specific ones that cannot be transferable to other layers, they have the possibilities to provide the transferable ones as the inter-layer representations for other layers to improve the link prediction performance. Hence, in this research, we aim to simultaneously utilize both inter-layer and intra-layer representations for link prediction in multi-layer networks.

To achieve this, the first challenge is to identify transferable ones in the intra-layer representations from different layers. Although these representations are difficult to track because of their dynamic and high-dimensional characteristics, they have the possibility to be detected and measured by reducing the differences of the representations of links corresponding to different layers. We attempt to introduce adversarial training to confuse the representations' layer sources to obtain the transferable ones. The second challenge is how to fuse the inter-layer and intra-layer representations for link prediction on a given layer. Since the structural correlations between different layers are changeable, it is difficult to solve this change in an unified manner. We try to reconstruct the gated unit to achieve the adaptive fusion between inter-layer and intra-layer representations during link prediction.

To address these challenges, we propose an end-to-end framework called Adversarial Enhanced Representation ($AER$) for link prediction in multi-layer networks. Our proposed framework consists of three main components: a representation generator, a layer discriminator, and a link predictor. The representation generator is proposed to learn the inter-layer and intra-layer representations of links, respectively. A gated unit in it is further reconstructed to achieve a adaptive fusion between inter-layer and intra-layer representations. This allows us to extract both inter-layer and intra-layer information from the network for graph embedding. The layer discriminator is designed to identify the layer to which a link belongs. It collaborates with the representation generator in an adversarial training process, aiming to better learn inter-layer transferable representations. Durng the minimax two-player game between the representation generator and layer discriminator, the representation generator attempts to learn inter-layer transferable representations to deceive the layer discriminator. In order not to be deceived, the layer discriminator attempts to accurately distinguish the layer sources of the learned inter-layer representations. Finally, the link predictor works in collaboration with the representation generator to predict whether a link is a missing link. By simultaneously utilizing inter-layer and intra-layer representations, the $AER$ exploits the multi-layer correlation to enhance the link prediction performance in multi-layer networks.

Our main contributions are as follows:

- To efficiently utilize the multi-layer correlation for link prediction, we propose a novel method that combines the inter-layer transferable representations with the intra-layer representations to improve the link prediction performances in multi-layer networks.

- To acquire inter-layer transferable representations, a minimax two-player game between the representation generator and layer discriminator is subtly designed via adversarial training.

- To enhance the links' representations for link prediction in multi-layer networks, we propose an efficient mechanism to achieve the adaptive fusion between inter-layer and intra-layer representations on a given layer.

- Extensive experiments on public datasets demonstrate that our method outperforms several state-of-the-art ones in link prediction in multi-layer networks. A simplified version of the codes have been published.

## 2 RELATED WORK

Link prediction in the graph-based networks is used to predict the possibility of a link between two nodes that have not yet been linked through known nodes, topology, and other information Liben-Nowell & Kleinberg (2007). With the depth of research, more and more scholars begin to focus on multi-layer networks Szell et al. (2010); Lee et al. (2015); Pujari & Kanawati (2015). Currently, existing link prediction methods for multi-layer networks are mainly divided into topology calculation

methods Najari et al. (2019); Abdolhosseini-Qomi et al. (2020); Luo et al. (2021) and deep learning methods Jalili et al. (2017); Mandal et al. (2018); Shan et al. (2020).

Topology calculation methods usually tend to calculate similarity scores between unlinked node pairs using network topology information. Najari et al. (2019) developed a link prediction framework that comprehensively considered the inter-layer similarity and features extracted from the prediction layer. In addition, Abdolhosseini-Qomi et al. (2020) proposed a layer reconstruction method, which utilizes the structural features of other layers for the optimal reconstruction of the target layer structure. Further, Luo et al. (2021) proposed a new multi-attribute decision making method which defines a layer similarity measure based on cosine similarity to achieve the weighting of each layer.

Deep learning methods formalize link prediction as a supervised binary classification problem, in which prediction models are trained according to the features of node pairs extracted or learned from the observed network structures. Specifically, Jalili et al. (2017) reported that, compared with naive Bayes and k-nearest neighbors, support vector machines provide better prediction performance. Mandal et al. (2018) reported that the quality of the feature group selection significantly influences the model's prediction effect. Shan et al. (2020) used a set of elaborate structural features of node pairs to feed a classification model and extracted complex structural features of node pairs from all layers for link prediction in multi-layer networks.

However, the methods above cannot achieve satisfactory link prediction performances in multi-layer networks because of their inefficient utilization of the multi-layer correlation. They have no ability to fuse the inter-layer representations and the intra-layer representations to improve the link prediction performance. The links among sharing nodes may belong to different layers, with each layer having intra-layer representations that are not sharable with other layers. Deep learning models tend to learn these intra-layer representations from links and may provide contradictory prediction results for links in different layers. Therefore, our research aims to learn inter-layer transferable representations to enrich intra-layer representations for link prediction in multi-layer networks.

## 3 PROBLEM DEFINITION

In this study, we aim to solve the the task of link prediction in multi-layer networks. To facilitate formulation, a multi-layer network with $K$ layers is denoted by $G = (g^1, g^2, \ldots, g^k, \ldots, g^K)$, where $K \geq 2$ and $g^k$ represents the $k$-th layer network. In addition, to design a universal method, we define $g^k = (V^k, E^k)$ as an unweighted and undirected graph, where $V^k$ and $E^k$ represent the existing node set and edge set in the $k$-th layer, respectively. In the $k$-th layer, some existent links are observed in $E^k$, whereas others are unobserved in $E_u^k$. We treat the different types of interactions in a complex reality network as different layers of the network. Each layer has the same set of nodes and a different set of edges, i.e., $E = \{E^1, E^2, \ldots, E^K\}$, $E_u = \{E_u^1, E_u^2, \ldots, E_u^K\}$. Formally, we define the link prediction problem in multi-layer networks as follows. Given $G = (g^1, g^2, \ldots, g^K)$, we design a matching set $Ms = \{\langle e, \delta \rangle \mid e \in E_u \cap \delta \in [0, 1]\}$, where each unobserved link $e$ in $E_u$ is assigned a reasonable value $\delta$ to quantify its existent likelihood. The link prediction can be a binary classification problem, which classifies unobserved links in $E_u$ into the missing links set $E_m$ and nonexistent links set $E_n$. The perfect solution to this problem is that $\delta = 1$ for unobserved existent links and $\delta = 0$ for nonexistent links.

## 4 METHODOLOGY

Our method aims to exploit the multi-layer correlation to improve the link prediction performance in multi-layer networks. To achieve this goal, the proposed $AER$ integrates a representation generator, a layer discriminator and a link predictor, as shown in Figure 1. First, the representation generator learns inter-layer transferable representations and intra-layer representations and fuses them together using a gated unit. Then, in order to avoid being fooled by the representation generator, the layer discriminator aims to correctly identify the layer source of learned inter-layer representations, which assists the representation generator to generate more realistic and effective inter-layer transferable representations. Finally, the link predictor is built on top of the representation generator and uses the fused enhanced representations of links to perform the primary link prediction task. The detailed descriptions of each component will be introduced in the following.

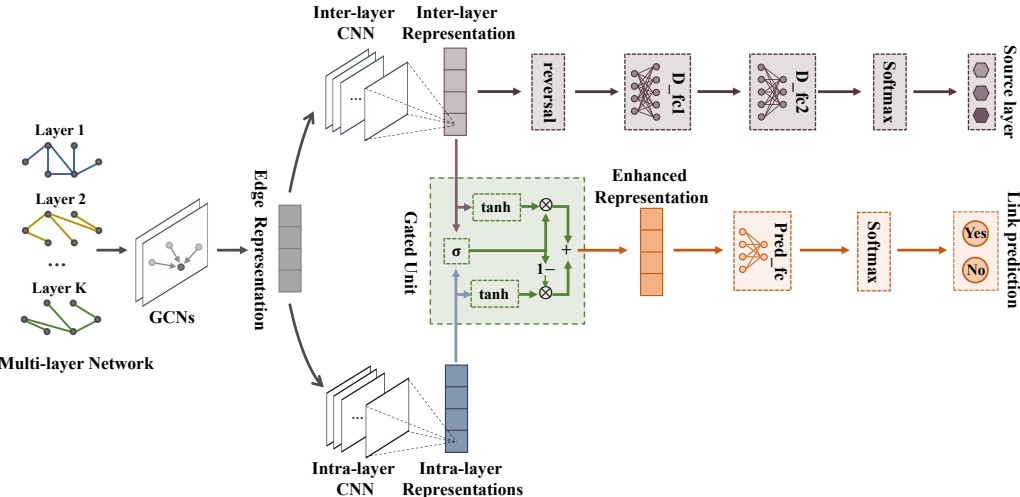

Figure 1: The overview of Adversarial Enhanced Representation ($AER$).

## 4.1 REPRESENTATION GENERATOR

Graph structured data can be processed and learned using different methods, including graph convolutional methods such as Wei et al. (2019) and node embedding methods such as Grover & Leskovec (2016). Among them, GCN is based on the linear combination of adjacency matrix and feature matrix to propagate the information, and the core idea is to use the edges to aggregate node information to generate new node representations. In this work, we use the GCNs to get the initial representation of nodes and edges. Limited by the attributes of multi-layer networks, there is not a simple and basic method to directly extract the representations from multi-layer graphs. Hence, we use multiple GCNs to separately extract the network structure information of each layer in multi-layer networks (i.e., there are $K$ GCNs attributed to a $K$-layer network graph).

Specifically, for each layer, we convert the network topology to the adjacency matrix $A^k$ and set the node ID to the initial matrix X. Then the pre-processing in the k-layer network can be denoted:

$$H^{k(l+1)} = \sigma(\widetilde{D^k}^{-\frac{1}{2}} \widetilde{A^k} \widetilde{D^k}^{-\frac{1}{2}} H^{k(l)} W^{k(l)}), \tag{1}$$

where $\widetilde{A^k} = A^k + I$, $I$ denotes the identity matrix, $\widetilde{D^k}$ denotes the degree matrix of $\widetilde{A^k}$, $W^k$ denotes the weight, $H^{k(0)} = X^k$, and $\sigma(\cdot)$ denotes the activation function. Each convolutional layer handles only first-order neighborhood information, and multi-order neighborhood information can be transferred by stacking several convolutional layers. In this work, we denote the number of convolutional layers $l$ as 2, in other words, $H^{k(1)} \in \mathcal{R}^{|V| \times d'}$ is the node representations $N^k$ processed by GCNs in the $k$-th network layer, where $d'$ denotes the dimension of the node representations. Then, the initial representation of a link $e$ in the $k$-th network layer can be denoted as:

$$IR_e^k = N_{n_l}^k \oplus N_{n_r}^k, \tag{2}$$

where $n_l$ and $n_r$ denote the left and right nodes of the link $e$ respectively, $\oplus$ denotes the concatenation operation, and the dimension of the edge representations can be denoted as $d = 2d'$. Therefore, $IR = \{IR^0, IR^1, \ldots, IR^K\}$ denotes the initial representations of edges in the multi-layer network.

Next, in order to learn fine-grained representations, we apply CNNs to further learn inter-layer transferable representations $TR$ and intra-layer representations $SR$. The filtering operation of consecutive $h$ edges starting from the i-th edge can be expressed as:

$$TR = \sigma(W_t \cdot IR_{i:i+h-1}), \tag{3}$$

where $\sigma(\cdot)$ denotes the activation function and $W_t$ denotes the weight of the convolution filter. Similarly, the intra-layer representations $SR$ is obtained:

$$SR = \sigma(W_s \cdot IR_{i:i+h-1}), \tag{4}$$

where $W_s$ denotes the weight of the corresponding convolution filter.

In this research, we aim to aggregate multi-layer information to improve link prediction performances. To achieve this goal, we introduce a gated unit from Arevalo et al. (2017) to fuse the inter-layer and intra-layer representations of links. The data flow diagram of this gated unit is shown in Figure 1. Formulally, the weight of the fusion operation is calculated by:

$$z = \sigma(W_z \cdot [SR \oplus TR]), \tag{5}$$

where $\oplus$ denotes the concatenation operation and $W_z$ denotes the associated learnable parameter matrix. Then, the final enhanced representations of links that are ultimately used to predict missing links in multi-layer networks are defined as $ER$:

$$ER = z * tanh(W_{zs} \cdot SR) + (1 - z) * tanh(W_{zt} \cdot TR), \tag{6}$$

where $W_{zs}$ and $W_{zt}$ denotes the associated parameter, and $tanh(\cdot)$ denotes the activation function.

We denote the representation generator as $M_r(G; \theta_r)$, where $G$ denotes the original multi-layer network and $\theta_r$ denotes all parameters in the representation generator. The generator passes $TR$ to the discriminator for cooperative learning to generate better and realistic inter-layer transferable representations, and the corresponding learning parameter involved is defined as $\theta_{r-}$. Meanwhile, the link predictor uses the final enhanced representations $ER$ from the generator to predict missing links in multi-layer networks, and they learn better parameters $\theta_r$ together.

## 4.2 LAYER DISCRIMINATOR

To learn inter-layer transferable representations of links in the multi-layer network, the first step is to identify the differences between representations at different layers. To measure this difference, we design the layer identifier, which is mainly composed of two fully connected layers and corresponding activation functions. We use $M_d(TR; \theta_d)$ to denote the layer discriminator, where $\theta_d$ denotes all parameter that the layer discriminator needs to learn.

Specifically, for a given link sample $e$, the purpose of this module is to identify which layer of the multi-layer network the link $e$ originates from based on the representation $TR_e$ passed in by the generator. The process is formulated as:

$$D_e = Softmax[W^{(1)T} \cdot ReLU(W^{(0)T} \cdot TR_e + b^{(0)}) + b^{(1)}], \tag{7}$$

where $W^{(0)} \in \mathcal{R}^{d \times h}$, $b^{(0)} \in \mathcal{R}^{h \times 1}$ and $ReLU(\cdot)$ denote the weight matrix, the bias vector and the activation function of the first fully connected layer respectively, $h$ denotes the dimension of the hidden layer, and $W^{(0)T}$ denotes the transpose of $W^{(0)}$. Similarly, $W^{(1)} \in \mathcal{R}^{h \times K}$, $b^{(1)} \in \mathcal{R}^{K \times 1}$ and $Softmax(\cdot)$ denote the weight matrix, the bias vector, and the activation function of the second fully connected layer, respectively. Then, we define the identification loss of the layer discriminator using cross entropy as follows:

$$Loss_d(\theta_{r-}, \theta_d) = -\frac{1}{|M|} \Sigma_{e \in M} [Y'_e \cdot log(D_e)], \tag{8}$$

where $M$ denotes the set of all link samples and $|M|$ denotes the number. $Y'_e$ denotes the ground layer label of the link sample $e$ in one-hot format which is a $K$-dimensional vector like $D_e$. For example, $[0, 1, 0]$ indicates that it originates from the 2-th layer in a 3-layer network.

To achieve the purpose of correctly identifying the layer sources of the learned inter-layer representations, the layer discriminator will adapt to reduce the loss $Loss_d(\theta_{r-}, \theta_d)$. When this loss is smaller, it indicates that the effect of the layer discriminator is better, that is, the fed $TR$ can help the discriminator better distinguish different layers. Conversely, the greater the loss, the more it reflects that the $TR$ given by the generator is non-identifiable and inter-layer transferable.

## 4.3 LINK PREDICTOR

The link predictor is built on top of the representation generator to accomplish our main task of link prediction. It is fed with the enhanced representations $ER$ given by the generator that incorporates the inter-layer and intra-layer representation, and passes through a fully connected layer to give the predicted result. We denote the link predictor as $M_p(ER; \theta_p)$, where $\theta_p$ denotes all learned parameters. For a given link sample $e$, the prediction process can be expressed as follows:

$$P_e = Softmax(W_p^T \cdot ER_e + b_p), \tag{9}$$

where $Softmax(\cdot)$ is used to normalize the predicted result, $W_p \in \mathcal{R}^{d \times 2}$ and $b_p \in \mathcal{R}^{2 \times 1}$ denote the weight matrix and the bias vector, respectively. Then, the predicted loss is defined as follows:

$$Loss_p(\theta_r, \theta_p) = -\frac{1}{|M|} \Sigma_{e \in M}[Y_e \cdot log(P_e)], \tag{10}$$

where $Y_e$ denotes the ground label of the link sample $e$ in one-hot format which is a 2-dimensional vector like $P_e$. Specifically, $[1, 0]$ indicates that the link is missing, and $[0, 1]$ indicates that the link exists. To better realize the prediction of missing links, the model seeks to minimize the prediction loss. The process of determining the optimal parameters can be expressed as follows:

$$(\hat{\theta}_r, \hat{\theta}_p) = \arg \min_{\theta_r, \theta_p} Loss_p(\theta_r, \theta_p). \tag{11}$$

## 4.4 MODEL INTEGRATION

In this section, we detail how the representation generator, the layer discriminator, and the link predictor work together for link prediction in multi-layer networks. On the one hand, the generator continuously generates inter-layer transferable representations so that the discriminator cannot correctly distinguish the layer sources of the inputted representations and strives to maximize $Loss_d(\theta_{r-}, \theta_d)$. On the other hand, in order not to be fooled by the generator, the discriminator strives to minimize $Loss_d(\theta_{r-}, \theta_d)$. A minimax game is formed between the representation generator and the layer discriminator as in Goodfellow et al. (2014), and the cooperation makes the model finally capture more realistic inter-layer transferable representations $TR$. Then, the representation generator cooperates with the link predictor to achieve the prediction task. The overall loss function is defined as follows:

$$Loss_{final}(\theta_r, \theta_d, \theta_p) = Loss_p(\theta_r, \theta_p) - Loss_d(\theta_{r-}, \theta_d). \tag{12}$$

Then, the parameter set selected by the model is the saddle point of the overall loss:

$$(\hat{\theta}_r, \hat{\theta}_p) = \arg \min_{\theta_r, \theta_p} Loss_{final}(\theta_r, \hat{\theta}_d, \theta_p), \tag{13}$$

$$\hat{\theta}_d = \arg \max_{\theta_d} Loss_{final}(\hat{\theta}_{r-}, \theta_d). \tag{14}$$

To achieve this goal, we add a gradient reversal layer which is introduced in Ganin & Lempitsky (2015) in the middle of the two modules. In the backpropagation process, the loss is automatically reversed before propagating to the representation generator. We use a stochastic gradient descent algorithm to optimize the model, and the parameter update process is as follows:

$$\theta_r = \theta_r - \eta(\frac{\partial Loss_p}{\partial \theta_r} - \frac{\partial Loss_d}{\partial \theta_r}), \tag{15}$$

where $\eta$ denotes the learning rate, which decays with iteration during the training stage:

$$\eta = \frac{\eta_0}{(1 + \alpha \times p)^\beta} \tag{16}$$

where $p$ denotes the ratio of the current iteration and the total iteration, $\eta_0 = 0.01$ denotes the initial learning rate. $\alpha = 10$ and $\beta = 0.75$ are hyperparameters, which are the same as Ganin & Lempitsky (2015). The detailed steps of the proposed method are summarized in Algorithm1.

## 5 EXPERIMENT

In this section, we will compare the performance of our proposed model with the state-of-the-art methods on five real-world datasets, where the specific statistics are shown in Table1.

## 5.1 DATASETS

To fairly evaluate the performance of the proposed $AER$, we consider the following five real multi-layer networks from the real world:

---

**Algorithm 1:** $AER$

---

  **Input:** A multi-layer network graph $G$ and the initial learning rate $\eta_0$.
  **Output:** The matching set of prediction results $Ms$.
**1** $Ms = \emptyset$;
**2** Obtain the tag set of the links $Y$ and the tag set of the layer sources $Y'$;
**3 for** *each iteration* **do**
**4**     Obtain the initial representations of nodes and links by Eq.1 and Eq.2 respectively;
**5**     Obtain the representations $TR$, $SR$ and $ER$ by Eq.3, Eq.4 and Eq.6 respectively;
**6**     Calculate $Loss_d$, $Loss_p$ and $Loss_{final}$ by Eq.8, Eq.10 and Eq.12 respectively;
**7**     Update the learning rate $\eta$ by Eq.16;
**8**     Update the parameters $\theta_p$, $\theta_r$ and $\theta_d$ by Eq.15;
**9 end**
**10 for** *each unobserved link $e$ in $E_u$* **do**
**11**     Calculate $P_e$ by Eq.9;
**12**     $Ms = Ms + (e, P_e)$;
**13 end**

---

Table 1: Statistics of multi-layer datasets.

| Dataset | #Layers | #Nodes | #Edges of different layers | | | | |
|---------|---------|--------|-----|-----|----|----|-----|
| $Aarhus$ | 5 | 61 | 193 | 124 | 21 | 88 | 194 |
| $Enron$ | 2 | 151 | 133 | 128 | - | - | - |
| $Kapferer$ | 4 | 39 | 158 | 223 | 76 | 95 | - |
| $Raion$ | 3 | 369 | 312 | 83 | 46 | - | - |
| $TF$ | 2 | 1564 | 14108 | 18471 | - | - | - |

- $Aarhus$ Magnani et al. (2013) comprises five relationships (Facebook, leisure, work, co-writing, and lunch) among the employees of the Aarhus computer science department.

- $Enron$ Tang et al. (2012) identifies the interaction information between employees. The two layers denote their superiors and colleagues relationship, respectively.

- $Kapferer$ De Domenico et al. (2014) is the interaction network of a tailor shop over a ten-month period, with four layers of the network representing the interaction of work, assistance, friendship and emotional, respectively.

- $Raion$ De Domenico et al. (2014) is a multi-layer network which denotes the railway stations in London, and the three layers of the network denote the stations connected by underground, above ground and $DLR$, respectively.

- $TF$ Jalili et al. (2017) is a multi-layer network formed by the collaboration of Twitter and Foursquare. The first layer identifies follow relationships on Twitter, and the second layer identifies friendship relationships on Foursquare.

The basic statistics of the five multi-layer network datasets are summarized in Table1. We treat observed links in the graph as positive samples and unobserved links as negative samples. For each dataset, we randomly split the dataset into training, validation and testing sets with 8:1:1 ratio.

### 5.2 COMPARISON METHODS

We include several traditional and state-of-the-art models in the comparison:

- $NSILR$ Yao et al. (2017) proposes a novel node similarity index based on layer relevance of multiplex network for link prediction by utilizing the intra- and inter-layer information.

- $MultiSup$ Shan et al. (2020) extracts the complex structural features of node pairs from all network layers to train classification models. In addition to a group of well-known similarity indicators such as CN, RA and Jaccard, two new features, the friendship between neighbors ($FoN$) and friendship among auxiliary layers ($Fal$), are designed.

- $MNERLP$ Mishra et al. (2022) calculates node relevance (local information) and edge relevance (global information) based on the summarized graph, and then combines both these factors to perform link prediction on unconnected pairs of nodes.

Table 2: Performance comparison between our method and the baselines on real-world datasets in terms of $Accuracy$ (%). Boldface scores indicate the best results. The underlined scores denote the second-best performance.

|  | *Aarhus* | *Enron* | *Kapferer* | *Raion* | *TF* |
|---|---|---|---|---|---|
| *NSILR* | 72.59±9.35 | 50.46±1.11 | 60.49±7.53 | 50.14±0.37 | 75.92±5.35 |
| *MultiSup* | 75.61±2.85 | 61.16±0.64 | 59.83±4.23 | 69.93±11.48 | 70.82±4.25 |
| *MNERLP* | 75.82±9.67 | 48.86±0.54 | 65.09±3.46 | 51.34±1.49 | 82.24±6.09 |
| *HOPLP* | 74.28±9.83 | 49.09±0.41 | 63.89±3.48 | 51.05±1.52 | 80.95±5.02 |
| *AER* | **76.53±2.65** | **77.46±6.96** | **71.83±7.05** | **79.96±2.04** | **82.52±4.03** |
| **Improv.** | 0.94% | 26.65% | 10.35% | 14.34% | 0.34% |

Table 3: Performance comparison between our method and the baselines on real-world datasets in terms of $AUC$ (%). Boldface scores indicate the best results. The underlined scores denote the second-best performance.

|  | *Aarhus* | *Enron* | *Kapferer* | *Raion* | *TF* |
|---|---|---|---|---|---|
| *NSILR* | 78.40±11.63 | 50.48±1.11 | 63.99±9.59 | 50.14±0.37 | 80.54±7.10 |
| *MultiSup* | 75.65±2.86 | 53.98±0.85 | 59.65±4.40 | 69.35±11.19 | 75.73±2.82 |
| *MNERLP* | 77.98±11.05 | 48.86±0.54 | 68.06±5.22 | 51.33±1.49 | 85.32±7.93 |
| *HOPLP* | 76.64±11.25 | 49.06±0.42 | 67.43±4.96 | 51.04±1.51 | 85.46±7.59 |
| *AER* | **82.78±2.82** | **88.96±2.76** | **77.17±4.33** | **86.67±1.60** | **88.60±0.45** |
| **Improv.** | 5.59% | 64.80% | 13.39% | 24.97% | 3.67% |

- $HOPLP$ Mishra et al. (2023) combines information from many layers into a single weighted static network while accounting for the relative density of the layers, and then calculates link likelihoods taking longer paths between nodes into account.

- $AER$ is our proposed model, which includes a representation generator, a layer discriminator and a link predictor. Further ablation study is provided in Section 5.6.

## 5.3 PARAMETER SETTINGS

In the representation generator, the dimension of hidden layer is set to 64 for all GCNs, $d'$ is 16, and $d$ is 32. Then, the dimensions of $TR$, $SR$ and $ER$ are consistent with $IR$ as 32. In the layer discriminator, the dimension of the hidden layer $h$ is 20, and the output size is the same as the number of layers $K$ in the network. In the link predictor, the output size is 2, such that the prediction results can be normalized. In addition, the number of iterations is set to 50, the batch size is 256, and the Adam algorithm is used for model optimization in this work. Furthermore, we use $python$ 3.7 and $pytorch$ 1.13 in implementation. For other baselines, we implement them with the published codes and tune their models based on the preferred parameter settings in the papers.

## 5.4 EVALUATION METRIC

We employ the following metrics to evaluate performance results: $Accuracy$ denotes the percentage of the number of samples predicted correctly in the total. $AUC$ (Area Under Curve) denotes the area under the $ROC$ (Receiver Operating Characteristic) curve. In general, the larger the above metrics are, the better the models perform.

## 5.5 PREDICTION RESULTS AND ANALYSIS

To investigate the performance of the proposed $AER$, we compare it with seven state-of-the-art methods on the five multi-layer network datasets. Table 2 and Table 3 respectively show the performance comparison of several models for link prediction in multi-layer networks in terms of $Accuracy$ and $AUC$. From the tables, we have the following observations: $AER$ substantially outperforms all the other baselines in terms of all metrics, verifying the effectiveness of our method. We attribute such significant improvements to the learning of fusing the inter-layer transferable representations and intra-layer representations, so as to learn the comprehensive representations of links in multi-layer networks effectively. Generally speaking, $MultiSup$ and $MNERLP$ achieve better performance than other baselines in most cases. It is reasonable since $MultiSup$ and $MNERLP$

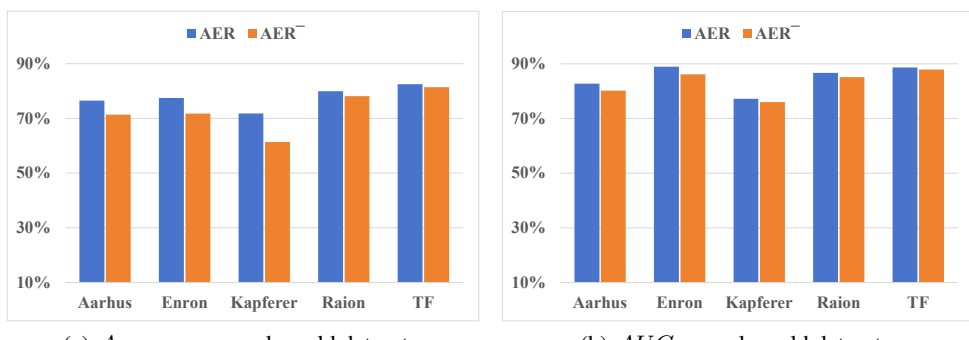

(a) *Accuracy* on real-world datasets.      (b) *AUC* on real-world datasets.

Figure 2: Ablation study for the effectiveness of our method.

also consider more comprehensive information in comparison. What is unexpected is the performance of $HOPLP$ on these real-world datasets. The reason for this may be that some information is lost when combining information from multiple layers into a single weighted static network.

## 5.6 ABLATION STUDY

To demonstrate the importance of adversarial training, we design a variant of $AER$ to compare its performance for link prediction in multi-layer networks. $AER^-$ is the variant model without the layer discriminator that learns only the intra-layer representations of links in a multi-layer network, but not the inter-layer transferable representations of links in a multi-layer network. As shown in Figure 2, the performance of $AER$ outperforms $AER^-$ in terms of all metrics. These results meet our expectations. This fully demonstrates the importance of fusing the inter-layer transferable and intra-layer representations, that is, the effectiveness of the minimax game between the representation generator and the layer discriminator.

## 5.7 CASE STUDY

We conduct an experiment to demonstrate the significance of fusing the inter-layer transferable and intra-layer representations of links for link prediction in multi-layer networks. To be specific, we choose the first layer network of the $Aarhus$ dataset as the given layer for performance comparisons with respect to whether or not to add the inter-layer transferable representations. When relying only on the topology information of this single network layer, the model performance is able to reach 0.66 and 0.78 in terms of $Accuracy$ and $AUC$, respectively. When adding other network layers as auxiliary network layers, combined with the inter-layer transferable representations, the model performance is able to improve to 0.71 and 0.83 in terms of $Accuracy$ and $AUC$, respectively. This case study once again demonstrates the effectiveness of fusing the inter-layer transferable representations and intra-layer representations of links in multi-layer networks. In particular, when a network layer is sparse and contains relatively little information, it is feasible to obtain auxiliary information from other network layers to enhance the representations of links on the given layer.

## 6 CONCLUSIONS AND FUTURE WORK

In this research, we exploit the multi-layer correlation to improve the link prediction performance, which accelerates the understanding of the link prediction issue in multi-layer networks. Our proposed method can effectively acquire the inter-layer transferable representations during a minimax two-player game, further using these representations to enrich the intra-layer representations to achieve an adaptive representation fusion. Extensive experiments on real-world datasets show that our proposed method outperforms state-of-the-art methods. Based on our existing study, many additional methods can be explored. One possibility is to extend our study into a directed or weighted network. To propose a universal method for generalized networks, we simply represented a network as an unweighted and undirected graph in this study. Another possibility is to exploit the attribute information of nodes and links, such as additional textual description. Our proposed method is a general framework for link prediction. The acquirement of inter-layer transferable representations and intra-layer representations can be easily designed for multi-modal situations.

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
