# OpenReview forum: "Adversarial enhanced representation for link prediction in multi-layer networks"
_ICLR.cc/2024/Conference — Submitted to ICLR 2024_

### Official Review · Reviewer_zmKs · 2023-10-29

**Soundness:** 1 poor
**Presentation:** 2 fair
**Contribution:** 1 poor
**Rating:** 1
**Confidence:** 4

**Summary:**

The paper proposes an adversarial model for link prediction in multi-layer networks where a fixed set of nodes is connected through a different set of edges in each layer. The architecture has an **encoder** (representation generator), composed by one GCN per (network) layer which output the respective intra-edge embeddings (IR). IR is fed to CNNs, one which generates an output (TR) for an **adversarial classifier** (layer discriminator) whose goal is to predict the source layer of the link, and another whose output (SR) is combined with TR through a gating mechanism in order to obtain a new edge embedding (ER). ER is fed to a **binary classifier**  (link predictor) to predict the existence of the link. The rationale is that the representation of edges should be consistent (not equal) across layers in order to leverage their correlations. The authors evaluate the proposed method (AER) on 5 multi-layer network datasets, showing that it outperforms four other baselines w.r.t. Accuracy and AUC. An ablation study shows the benefits of incorporating inter-layer correlations using TR.

**Strengths:**

S1. The paper considers an interesting adversarial setting to learn edge representations that are sufficiently "layer-invariant" to mislead a discriminator but still useful to perform intra-layer link prediction.

**Weaknesses:**

W1. The relationship between the present work with the body of works on link prediction in heterogeneous graphs (multiple edge types) is never established in the paper. As a result, GNN-based models for heterogeneous graphs (e.g., R-GCN, HIN) were never included in the comparison. It is worth mentioning that these models do capture correlations between different edge types, which are arguably analogous to edges in different layers.

W2. The problem tackled in the paper is also likely related to embeddings for knowledge graphs (e.g., DGL-KE). This is also missing in the related work and in the experimental section.

W3. The combination of an adversarial loss with a binary classification loss (three sets of parameters to be optimized) can pose several challenges to the training, requiring a delicate balance between the two losses. Yet, there is no hyperparameter in $Loss_{final}$ (Eq. 12) to control this balance. There are no plots showing the evolution of the loss components.

W4. No study or empirical results regarding the method scalability. Networks are relatively small.

W5. It is hard to determine whether the comparison with the baselines is fair.

W6. Some design choices in the proposed method are not well-justified. Additional experiments need to be included in the ablation study.

W7. Notation and typesetting need improvement. Spelling must be reviewed.

W8. The results are not reproducible based on the information provided in the submission. Appendix was not used.

**Questions:**

Q1. What is the difference between link prediction in multi-layer networks and in heterogeneous networks (multiple edge types)? Why the methods proposed for the latter were not discussed?

Q2. Same question as before, but considering knowledge graphs.

Q3. How do you determine that the model has converged? When it does, what is the discriminator accuracy? 1/K? What to the loss curves look like during the optimization? What are typical values of z in Eq. (6) or, in other words, how much of TR is relevant for link prediction?

Q4. How does the model training and inference time scale with the number of nodes, edges and layers? What were the training times obtained for each dataset?

Q5. Do the baselines have roughly the same number of parameters? Do they require about the same compute power for training? In other words, could the performance gains be coming at the cost of higher computing needs?

Q6. Some issues have to do with disregarding invariances:
- Since Eq. (2) is based on concatenation, does it consider both (i,j) and (j,i) when learning the IR for an edge e=(i,j)?
- Why use a CNN to transform IR? Does Eq. (13) yield TR for edge i? The CNN isn't invariant to the order of the edge sequence. Shouldn't this be a problem?
- Why gating instead of a concatenation followed by a MLP? What is the benefit of this specific choice?
- Why softmax for link prediction? Use sigmoid and reduce a few parameters.

Q7. Consider:
- Not using 'dot' for matrix multiplication
- Using 'odot' for element-wise multiplication
- Correctly typesetting log, tanh, and multi-letter variables

---

### Official Review · Reviewer_rv4f · 2023-10-29

**Soundness:** 3 good
**Presentation:** 3 good
**Contribution:** 1 poor
**Rating:** 3
**Confidence:** 5

**Summary:**

This paper notices that some real-world scenarios can be represented as a multi-layer network where different layers share the same set of vertices but different set of edge connections. This paper proposes a representation generator, a layer discriminator, and a link predictor to effectively integrate both intra-layer and inter-layer information. Experiments on 5 datasets are conducted to evaluate the performance of the proposed model.

**Strengths:**

1. Multi-layer network is a variant of existing single-layer network and is worth researching. This paper identifies an interesting problem and designs a working model to solve it.

2. Specifically, although the desisn of a representation generator is not new, the proposed layer discriminator looks interesting in the multi-layer setting.

3. Overall, the writing of this paper is clear enough for readers to understand the concept.

**Weaknesses:**

1. Missing baseline models. Although this paper proposes an interesting research question and a promising model architecture, it fails to mention two highly related works, MANE [1] and MLHNE [2]. These two works both work on multi-layer network embedding and both propose intra-layer and inter-layer concept to integrate information of different network layers. This submitted work lacks the discussion and comparison with these two papers.

[1] Li, J., Chen, C., Tong, H., & Liu, H. (2018, May). Multi-layered network embedding. In Proceedings of the 2018 SIAM International Conference on Data Mining (pp. 684-692). Society for Industrial and Applied Mathematics.

[2] Zhang, D. C., & Lauw, H. W. (2021). Representation Learning on Multi-layered Heterogeneous Network. In Machine Learning and Knowledge Discovery in Databases. Research Track: European Conference, ECML PKDD 2021, Bilbao, Spain, September 13–17, 2021, Proceedings, Part II 21 (pp. 399-416). Springer International Publishing.

2. Insufficient experiment tasks. For network embedding area, if the proposed model is unsupervised,  both link prediction and node classification are important evaluation tasks. Node classification has been a standard evaluation task in network embedding area and has been used in various papers, including the above mentioned two missing papers, but this submitted paper contains link prediction but not node classification.

3. Small datasets. The datasets in the paper contain <2K vertices, which is much smaller than real-world scenarios. Thus it's difficult to say if the proposed model can scale to large networks efficiently. I expect to see experiments on larger datasets, say 100K vertices.

**Questions:**

1. Why is standard deviation at Figure 2 missing?

---

### Official Review · Reviewer_m29U · 2023-10-30

**Soundness:** 2 fair
**Presentation:** 1 poor
**Contribution:** 2 fair
**Rating:** 3
**Confidence:** 5

**Summary:**

The paper proposes a multi-layer GNN for link prediction, which consists of representation generator, layer discriminator, and a link predictor. The representation generator includes GCNs for different layers for initial node embeddings, followed by CNNs to get both inter- and intra- layer node embeddings. The layer discriminator aims to identify the layer sources of learned inter-layer representations. The experimental results show some improvement over some baselines and ablation studies demonstrate the necessity of layer discriminator.

**Strengths:**

1. The experimental results show some improvement over some baselines and ablation studies demonstrate the necessity of layer discriminator.

**Weaknesses:**

1. Multi-layer GNNs or multiplex GNNs are not new topics, but the paper does not comprehensively review the existing literature in this topic. In addition to the literature review, the authors did not include these works as baselines for experimental comparisons. For example, existing related works include:
[1] Mitra, Anasua, et al. "Semi-supervised deep learning for multiplex networks." Proceedings of the 27th ACM SIGKDD conference on knowledge discovery & data mining. 2021.
[2] Zhang, Weifeng, et al. "Multiplex graph neural networks for multi-behavior recommendation." Proceedings of the 29th ACM international conference on information & knowledge management. 2020.
[3] Li, Jundong, et al. "Multi-layered network embedding." Proceedings of the 2018 SIAM International Conference on Data Mining. Society for Industrial and Applied Mathematics, 2018.

2. The proposed approach is not well-motivated and the writings need a lot of improvements. For example, it is not clear why CNNs are applied after GCNs for inter- and intra-layer node embeddings. In addition, it looks to me that layer discriminator and link predictor serve for a similar purpose, but the uniqueness of these components is not further highlighted.

3. The datasets used in this work are quite small. The largest dataset only contains ~1500 nodes. The authors are suggested to use the datasets from the above mentioned works to conduct more comprehensive experimental evaluations.

4. Minor: The notations are not very readable. Many notations are named a bit long (e.g., IR, TR, SR, etc.).

**Questions:**

1. Why CNNs are applied after GCNs for inter- and intra-layer node embeddings?
2. What are the differences between layer discriminator and link predictor, since both of them aims to predict link existence?

---

### Official Review · Reviewer_L4YV · 2023-11-01

**Soundness:** 2 fair
**Presentation:** 2 fair
**Contribution:** 2 fair
**Rating:** 3
**Confidence:** 4

**Summary:**

This paper proposes a novel framework called Adversarial Enhanced Representation (i.e., AER) for link prediction in multi-layer networks. AER is composed of three components: a representation generator, a layer discriminator, and a link predictor. The representation generator can simultaneously utilize both inter-layer and intra-layer representations in multi-layer networks. Moreover, AER can effectively acquire the inter-layer transferable representations to enrich the intra-layer representations via adversarial training between the representation generator and layer discriminator. Extensive experiments on real-world datasets show the methods’ effectiveness.

**Strengths:**

S1: The paper is well organized and easy to understand.
S2: The method as a framework can be generalized easily to other tasks on the multi-layer networks.
S3: Extensive experiments on real-world datasets show the propsed method can effectively and efficiently capture the information between the different layers in the multi-layer networks.

**Weaknesses:**

W1: The novelty of AER is insufficient. Many of the components used in AER are traditional and widely known models or operators (e.g., GCN, CNN). Authors do not explain clearly about the motivation for combining them. And in my opions, many of the operations represented in this paper seem to be basic to existing methods.
W2: The experiment is insufficient and the baslines maybe lacking. First, the old-fashioned link prediction baselines are missing, including but not limited to common neighbors (CN), Jaccard, preferential attachment (PA), Adamic-Adar (AA), resource allocation (RA), Katz, PageRank (PR), and SimRank (SR). (2) Some SOTA models are also needed such as SEAL (Link Prediction Based on Graph Neural Networks).
W3: The writing of the paper needs to be improved. In particular, the third paragraph of the introduction is not well understood and lacks citations to support points. There is no need to introduce too many details of the experimental parameters in the methods section. And the description in the algorithm section is rather redundant and does not fit the form of the algorithm.

**Questions:**

Q1: What is the mutli-layer network introducted in the intro? Could the authors provide a more specific description of such datasets. Also, are they any different from the graph-level datasets used in GNN research?
Please refer to weaknesses for other questions.

---

### Meta-Review · Area_Chair_ANow · 2023-12-11

**Metareview:**

This paper studies link prediction in multi-layer networks and proposes a novel graph embedding method called Adversarial Enhanced Representation (AER). AER consists of a representation generator, a layer discriminator, and a link predictor, aiming to learn and fuse inter-layer and intra-layer representations while deceiving the layer discriminator through inter-layer transferable representations. There are some weaknesses of the paper raised from the review comments and discussions, including the methodology novelty, the experiment sufficiency, and the presentation. As the authors provided no feedback, the concerns raised are still unaddressed.

**Justification For Why Not Higher Score:**

There are some weaknesses of the paper raised from the review comments and discussions, including the methodology novelty, the experiment sufficiency, and the presentation. As the authors provided no feedback, the concerns raised are still unaddressed.

**Justification For Why Not Lower Score:**

N/A

---

### Decision · Program_Chairs · 2024-01-16

Reject